# Analyzing and Forecasting Energy Consumption in China's Manufacturing Industry and Its Subindustries

**Wei Sun** [1,*] **, Yufei Hou** [2] **and Lanjiang Guo** [3]

1   China Institute of Manufacturing Development, Collaborative Innovation Center on Forecast and Evaluation of Meteorological Disasters, Nanjing University of Information Science & Technology, Nanjing 210044, China

2   Business School, Nanjing University of Information Science &Technology, Nanjing 210044, China; 20171213558@nuist.edu.cn

3   Wells Capital Management, Wells Fargo Bank, Boston, MA 02116, USA; lguo@wellsfargo.com

*   Correspondence: sunwei16@nuist.edu.cn

**Abstract:** In the context of new industrialization, the energy problem being experienced by the manufacturing industry has aroused social concerns. This paper focuses on the energy use of 27 subindustries in China's manufacturing industry and it develops an energy consumption index for 1994–2015. Subsequently, the method of grey relational analysis is used, with the full period divided according to years in which change points occur. The empirical analysis indicates that the energy consumption indexes generally exhibit a declining trend. Using the grey model (GM (1,1)) to forecast the index indicates a continued downward trend up to 2025 for energy-intensive industries, which is a more optimistic scenario than the trend forecast for the whole manufacturing sector. Thus, these energy-intensive industries do not drag down the performance of the whole manufacturing industry in regard to energy intensity. In future, more attention should be paid to energy-saving efforts by nontraditional high-energy-consuming industries. Although the results show that energy efficiency is improving in China, total annual consumption is rising rapidly. Therefore, the industry needs to continue to strengthen independent innovation and improve the efficiency of new energy use. The Chinese government should formulate feasible long-term plans to encourage enterprises to save energy.

**Keywords:** manufacturing industry; energy consumption; technological innovation; trend forecast; energy-intensive industries

## 1. Introduction

The fruitful achievements of the Third Scientific and Technological Revolution have made people's lives easier and promoted technological innovation and structural transformation in traditional industries. The manufacturing industry is a crucial industry and a key indicator of the economic level of a country or region. Many developed countries already possess advanced manufacturing sectors, but they continue to seek new opportunities and to reform their manufacturing industries to ensure an invincible position in the face of modernization and technological development. A typical example is Germany, as its "Industry 4.0" focuses on intelligent development, which includes an emphasis on the efficiency of outputs, material usage, and energy consumption [1]. The focus of this paper is the energy consumption of the manufacturing sector in China.

When compared with developed countries, China's manufacturing industry started late. Although China has made remarkable achievements in regard to its economic development, to achieve

such rapid results, the manufacturing industry adopted an extensive development mode and consumed large quantities of nonrenewable energy in the early stage of development. This has resulted in negative impacts on China's sustainable development. Today, China has become the largest energy consumer in the world [2], and the problems of energy shortages and low efficiency have also aroused widespread concern in Chinese society. The problem is reflected in Made in China 2025, the major strategy for China's manufacturing industry.

This paper evaluates the efficiency of energy use by the subindustries in China's manufacturing industry from 1994 to 2015. To do so, we calculate the energy consumption index of each industry in each year by using the method of maximum deviation, which can ensure that the weights assigned to each index are objective. Afterwards, we determine the years of change points in each industry using the method of grey relational analysis. First, the impacts of China's energy policies on the manufacturing industry's energy consumption index and change points are thoroughly analyzed. Subsequently, we use the grey model (GM) (1, 1) grey model to forecast the energy use of China's manufacturing industry. The GM (1, 1) model quantifies the concept of system information sampling, conceptualizes the quantitative model, and optimizes the model to predict some unknown data. The purpose of this paper is to determine the specific problems regarding energy use in China's manufacturing industry through a comprehensive longitudinal analysis of the industry's energy consumption and to make suggestions for China's sustainable development. We consider that this will provide a basis for the government and the manufacturing industry to develop a clearer understanding of the manufacturing industry, which can assist departments to formulate policies that achieve better results in practice. Therefore, the study of energy use is significant, not only theoretically, but also from a practical perspective.

The rest of the paper is organized as follows. Section 2 provides a literature review. Section 3 describes the method of maximum deviation and the grey relational analysis. Section 4 introduces the indicators and data and demonstrates the empirical analysis process using a key subindustry as an example. Section 5 analyzes the energy consumption index of different subindustries and explains the characteristics of the annual rate of growth of the index. Section 6 forecasts the index trend up to 2025. Section 7 concludes the paper.

## 2. Literature Review

Energy consumption has attracted widespread attention from international scholars. Studies on energy use by industry can be divided into three categories, which we discuss in turn below: research exploring the relationship between the economy and energy use; research evaluating the regional industrial energy efficiency; and, research exploring how to improve the industrial energy efficiency through environmental regulation and technological innovation.

### 2.1. Energy Use and the Economy

Many studies have confirmed that improving energy efficiency was of great significance to economic growth [3–5]. Arbex and Perobelli [6] integrated two models to analyze the impacts of economic growth on the consumption of energy. The empirical tests in Feng's paper indicated that there was a unidirectional causality running between energy intensity and economic growth [7]. Nasreen and Anwar [8] explored the relationship between economic growth, trade, and energy in Asia, and found that economic growth and trade openness had a positive impact on energy use. While many scholars have confirmed the one-way relationship between economy and energy [9–11], others have provided evidence that there is a two-way relationship between them [12,13].

Some scholars have explored the relationship between green economy and population, with results showing that a green economy was beneficial to employment in most countries [14,15]. Lombardi and Schwabe [16] focused on the benefits of battery storage systems under a shared economy model. Although the results of these literatures were valuable, the drawback was that the discussion was relatively macroscopic, and there was a lack of specific industrial empirical research.

## 2.2. Evaluating Industrial Energy Efficiency

Because of the intricate relationship between the economy and energy, a comprehensive understanding of the energy utilization of a region or an industry is particularly important. There is significant literature evaluating the energy consumption of industry. Zha [17] used data on 36 industry subsectors in China from 1993 to 2003 to evaluate energy consumption. By using improved index decomposition methods, the intensity and structural effects of different industries were analyzed. The results showed that the electrics subindustry performed well on energy consumption during this period, whereas the production and supply of gas subindustry and the petroleum processing and coking subindustry performed poorly. Wang et al. [18] also used index decomposition analysis to study the drivers of change in energy consumption and energy-related emissions, but their research focused more on the comparison between this method and structural decomposition analysis. One paper focused on evaluation methods and evaluated high-energy industries [19]. Based on our review of the above articles, we determined that different results could be obtained using different evaluation methods, making the choice of index and method very important in the evaluation of energy consumption.

Cahill and Brian [20] combined national energy statistics with company-level figures to quantify the energy savings that are achieved by companies, and assessed the effectiveness of a national energy-saving program. Based on defined system boundaries, Rojas-Cardenas [21] estimated the energy intensity of the Mexican iron and steel industry in 2010, comparing it with the United States and China, and found that the Mexican industry was more energy efficient. Scholars from Sweden, Japan, and Colombia have conducted similar energy efficiency studies [22–24]. Another study analyzed energy use, examining energy use flows for 186 economies worldwide using systems input–output analysis [25].

Regarding energy options, Urge-Vorsatz et al. [26] proposed a framework considering the indirect costs and benefits that were not included in previous studies to provide a key method to assess the multiple impacts of energy options.

## 2.3. Exploring How to Improve the Industrial Energy Efficiency

### 2.3.1. Improving Energy Consumption and Environmental Regulation

The conventional view within the existing literature is that the main means of influencing the energy consumption of industry are environmental regulation and technological innovation. Nilsson's paper [27] examined in detail how the transportation sector can achieve better development in the green energy economy and affirmed the power of governance. Lin and Chen [28] predicted the energy demand of China's manufacturing industry through scenario scoring and suggested that timely and effective government measures were conducive to decreasing energy consumption. Yuan [29] discovered that environmental regulations could assist in improving the energy performance of the manufacturing sector by building an extended concept data model. In Hou's paper, the green innovation growth rate of 28 manufacturing industries in China was deduced, based on the slacks-based measure-directional distance function method. The results indicated that the overall level of green innovation growth in China's manufacturing was relatively low, and that environmental regulation to promote green technology innovation was not being realized in the manufacturing industry [30]. Chen and Gong used panel data on manufacturing and piecewise linear utility functions to construct a data envelope analysis model that involved energy consumption and environmental regulations. Their research showed that low-energy-efficiency policies were conducive to the development of high energy-consuming industries [31]. Some scholars considered that the best measures for energy conservation and emission reductions did not require innovative technology, as good government policies and regulations were sufficient to achieve the energy goals [32].

Other scholars have examined how to ensure the practical application of national energy policy. In general, they considered that more ambitious energy efficiency policies should be promoted [33] and

that attention should be paid to balancing stakeholder interests [34,35] to accelerate the transformation of the green energy economy [36].

2.3.2. Improving Energy Consumption and Technological Innovation

With regard to technological innovation, many scholars have focused on intelligent manufacturing. From the viewpoints of theory and policy, Zhou provided an introduction to the digitalization and "intelligentization" of the manufacturing industry in China, including discussing how to upgrade mechanical systems, such as injection molding machines, to reduce energy consumption [37]. Haeffner and Panuwatwanich analyzed the perceived impacts of "Industry 4.0" on Germany's manufacturing industry, believing that the development of intelligent manufacturing could lead to energy efficiency [1]. To achieve innovation in the manufacturing industry, Liang and Huang considered that the manufacturing industry needed to integrate well with the service industry [38].

More and more scholars have focused on improving the energy innovation system. At the beginning of the 21st century, Sager et al. [39] attempted to strengthen the understanding of the energy innovation system and to enhance the world's recognition of this concept. Ju et al. [40] conducted in-depth research and proposed a hybrid energy system that evaluated operational performance in regard to energy, the economy and the environment. National energy policies were also considered to be important for sustainable energy systems [41]. Some scholars studied how the theoretical framework of the business model innovation of energy cooperation could assist in meeting realistic challenges [42]. Other studies showed that economic opportunities for innovation can be created by adjusting fossil fuel prices, but that this policy was not sufficient to achieve the goal of energy saving and emission reductions [43,44]. Others were concerned about the development and utilization of new energy sources. The assessment of national renewable energy utilization [45–47] was a popular topic in recent years. It is important that, in the process of technological innovation, we pay attention to whether innovations are reasonable because some so-called innovative clean technologies are associated with high risks and a low rate of return [48], and thus have no practical significance.

The above literature affirms the importance of energy issues to social development. However, based on our detailed examination of the literature, we find that there are several problems and gaps:

- There are few articles evaluating comprehensive energy use in China's manufacturing industry. Any such literature tends to be based on outdated data from the beginning of the 21st century.
- There are fewer articles on energy consumption in the subindustries of China's manufacturing industry.
- When evaluating regional energy consumption, the method used to weight indicators is subjective.
- Scholars have mostly been concerned with total energy consumption. Even though energy intensity has been studied, little attention has been paid to the changing characteristics of the annual growth rate.
- While research has been conducted regard the historical energy situation, there is no agreed empirical method for forecasting the future.

This paper attempts to fill some of these gaps. To ensure objectivity, it is reasonable to use the method of maximizing deviations to assign weights to the energy consumption indexes of manufacturing industries. Furthermore, for segmental evaluation, we use the grey relational analysis to determine the index change points to comprehensively evaluate the latest developments in various sectors of China's manufacturing industry. Based on the above results, we further explore the characteristics of the annual rate of growth of the energy consumption index to determine the relationship between the energy consumption index and energy policy. Subsequently, we forecast the energy consumption index of China's manufacturing industry for 10 years, with the aim that our discovery of the forecast process will inspire the sustainable development of manufacturing industry.

## 3. Methods

In this section, we will introduce three analytical methods, the maximum deviation method, the grey relational analysis to determine change points and the GM (1, 1) forecasting model.

### 3.1. Maximum Deviation

The evaluation of energy consumption in the manufacturing industry is a multi-attribute decision problem. The method of maximum deviation can be used to objectively determine the index weight, the allocation of which can affect the rationality of the indexes [49]. Xu [50] and Zhang et al. [51] highlighted the maximum deviation method for uncertain multi-attribute decision making. The algorithm is simple, clear in concept, and easy to implement using Microsoft Excel. The biggest advantage of the method is that it excludes subjective arbitrariness, ensuring that the evaluation results are objective and reliable. It provides a new way to solve the multi-attribute decision problem where the weight information on attributes is uncertain and the attribute value is given in the form of interval number [50]. Many scholars have adopted this method in their research. For example, Chen et al. [52] used the maximum deviation method to determine a ranking order for logistics suppliers and Lo and Guo [53] demonstrated the feasibility and practicability of the method using the example of a hospital maintenance quality audit. The research results confirm that the method is scientific and efficient [49–53].

There are four possible types of indicators, namely cost, efficiency, range, and fixed indicators [54]. In this paper, only the cost indicator is used, based on the "the smaller, the better" principle.

Let $A = \{A_1, A_2, \ldots, A_n\}$ be the multiple attribute decision-making scheme, $G = \{G_1, G_2, \ldots, G_3\}$ is the index set and $y_{ij}(i = 1, 2, \ldots, n; j = 1, 2, \ldots, m\}$ is the index value of the scheme $A_i$ to $G_j$ index. The decision matrix is $Y = (y_{ij})_{n \times m}$.

To eliminate incomparability, we need to use a dimensionless process for the evaluation index. As noted, this paper only involves the cost indicator. Such dimensionless indicators can be processed, as follows:

$$Z_{ij} = \frac{y^{max} - y_{ij}}{y^{max} - y^{min}}, \ i = 1, 2, \ldots, m. \tag{1}$$

After dimensionless processing, the decision matrix $Z = (Z_{ij})_{n \times m}$ can be obtained. The larger is parameter $Z_{ij}$, the better. We assume that the weighted vector of the evaluation index is $w = \{w_1, w_2, \ldots, w_m\}^T$. Based on the weighted vector $w$, a normalized weighted decision matrix can be constructed. The multi-index comprehensive value of decision scheme $A_i$ can be obtained using the simple additive weighted method:

$$Di(w) = \sum_{j=1}^{m} z_{ij} w_{ij}, i = 1, 2, 3 \ldots n. \tag{2}$$

$D_i(w)$ is similar to the decision matrix $Z = (Z_{ij})_{n \times m}$, meaning that a greater $D_i(w)$ indicates a better scheme.

According to the influence of the final evaluation value of a certain index $G_j$ on the decision-making scheme $A_i$, the obtained weight can be determined. For indicator $G_j$, the dispersion of the program and other decision plans is $v_{ij}(w)$.

Let:

$$v_j(w) = \sum_{i=1}^{n} v_{ij}(w) = \sum_{i=1}^{n} \sum_{k=1}^{n} \left| z_{ij} - z_{kj} \right| w_j, \quad j = 1, 2, \ldots, m. \tag{3}$$

where $v_j(w)$ is the sum of deviations of each decision scheme when compared with others under index $G_j$. According to the above analysis, the weighting vector $w$ should achieve a maximum; therefore, the objective function can be formulated, as follows:

$$\max F(W) = \sum_{j=1}^{m} v_j(W) = \sum_{j=1}^{m} \sum_{i=1}^{n} \sum_{k=1}^{n} \left| z_{ij} - z_{kj} \right| W_j \tag{4}$$

The above-mentioned weighted objective function is equivalent to the following nonlinear programming problem:

$$\begin{cases} max \ F(W) = \sum_{j=1}^{m} v_j(W) = \sum_{j=1}^{m} \sum_{i=1}^{n} \sum_{k=1}^{n} \left| z_{ij} - z_{kj} \right| W_j \\ s.t. \ \sum_{j=1}^{m} w_j^2 = 1 \end{cases} \tag{5}$$

The solution for Equation (5) is as follows:

$$w_j^* = \frac{\sum_{i=1}^{n} \sum_{k=1}^{n} \left| z_{ij} - z_{kj} \right|}{\sqrt{\sum_{j=1}^{m} \left( \sum_{i=1}^{n} \sum_{k=1}^{n} \left| z_{ij} - z_{kj} \right| \right)^2}}, \ j = 1, 2, \ldots, m. \tag{6}$$

It can be proven that $w^* = (w_1^*, w_2^*, \ldots, w_m^*)^T$ is the only maximum point of the objective function $F(w)$. The traditional weighted vector generally meets the normalized constraints rather than the unit constraint; therefore, the unit weight vector $w^*$ is normalized:

$$\widetilde{w}_j^* = w_j^* / \sum_{j=1}^{m} w_j^* \tag{7}$$

Equation (7) is equivalent to:

$$w_j^* = \frac{\sum_{i=1}^{n} \sum_{k=1}^{n} \left| z_{ij} - z_{kj} \right|}{\sum_{j=1}^{m} \sum_{i=1}^{n} \sum_{k=1}^{n} \left| z_{ij} - z_{kj} \right|}, \ j = 1, 2, \ldots, m. \tag{8}$$

*3.2. Grey Relational Analysis of the Change Point*

Grey relational analysis is a well-established analytical method, although the search for change points is a small branch of it that has not been widely used. We use this method because we need to segment a time series, and the grey relational analysis of searching for change points is suitable for this purpose. We convert the calculated values into the form of a line graph. We consider that there may be an "inflection point" somewhere in the time series. If the proposed method calculates a change point exactly at this expected inflection point, it can indicate that the method is consistent with the actual situation and we can prove our conjecture. The method makes segmenting data columns very simple and it can also be implemented in Microsoft Excel and a software called GTMS 7.0 (Grey Theory Modeling System 7.0). Next, we explain the operation process of this method. Assuming that $X_0 = \left( x_{0(1)}, x_{0(2)}, \ldots, x_{0(n)} \right)$ is the sequence of system characteristics, $X_i = \left( x_{0(1)}, x_{0(2)}, \ldots, x_{0(n)} \right), i = 1, 2, 3 \ldots, m$, is the sequence of related factors. Let the sequence,

$X_i = \left( x_{0(1)}, x_{0(2)}, \ldots, x_{0(n)} \right), i = 1, 2, 3 \ldots, m$, be the corresponding fixed sequence. For $\varepsilon \in (0, 1)$, we assume that:

$$\gamma \left( x_{0(k)}, x_{i(k)} \right) = \frac{\min\limits_{i} \min\limits_{k} \left| x_{0(k)} - x_{i(k)} \right| + \delta \max\limits_{i} \max\limits_{k} \left| x_{0(k)} - x_{i(k)} \right|}{\left| x_{0(k)} - x_{i(k)} \right| + \delta \max\limits_{i} \max\limits_{k} \left| x_{0(k)} - x_{i(k)} \right|} \tag{9}$$

$$\gamma(X_0, X_i) = \frac{1}{n} \sum_{k=1}^{n} \gamma \left( x_{0(k)}, x_{1(k)} \right) \tag{10}$$

where $\varepsilon \in (0, 1)$ is the distinguished coefficient and $\gamma(X_0, X_i)$ is the grey relational grade between $X_0$ and $X_i$ $(i = 1, 2, \ldots, m)$. $\gamma(X_0, X_i)$, and the relational coefficients $\gamma \left( x_{0(k)}, x_{i(k)} \right)$ can be abbreviated as $\gamma_{0i}$ and $\gamma_{0i(k)}$, respectively.

The grey relational analysis algorithm search for change points can be carried out, as follows:

(a) Build the reference sequence. Let $X_0 = \left( x_{(1)}, x_{(2)}, \ldots, x_{(T)} \right), 5 \leq Ts \leq T \leq T\varepsilon \leq [n/2]$ be the reference sequence for the time series $X = \left( x_{0(1)}, x_{0(2)}, \ldots, x_{0(n)} \right), n \geq 10$, which is the first part of the series. Subsequently, $T_s \leq T_\varepsilon$ ($T_s$ and $T_\varepsilon$ are the years of the change points), where $T_s$, $T$ and $T_\varepsilon$ are integers.

(b) Build the comparative sequence, which is constructed as follows:

$$X_i = \left\{ x_{(T+i)}, x_{(T+i+1)}, \ldots, x_{(T+i+T-1)}, \ i = 1, 2, \ldots, n - 2T + 1 \right\} \tag{11}$$

where Equation (3) defines the comparative sequence with an order of $n - 2T + 1$.

(c) Calculate the relational grade $r_{1(T)}, r_{2(T)}, \ldots, r_{(n-2T+1)\ (T)}$ using $X_0$ and $X_1, X_2, \ldots, X_{n-2T+1}$, respectively, and determine the arithmetic mean $r_{(T)} = \frac{1}{n-2T+1} \sum\limits_{i=1}^{n-2T+1} r_{i(T)}$ of these grades of association $r_{(T)}, 5 \leq Ts \leq T \leq T\varepsilon \leq [n/2]$, which is referred to as T's overall relational grade.

(d) Determine the change point:

$$\eta(T) = \left| \frac{r(T+1) - r(T)}{r(T)} \right| \times 100, \ T = (T_s, T_{s+1}, \ldots, T_{\varepsilon-1}, \eta(T_\varepsilon) = 0 \tag{12}$$

where $\eta(T)$ and $5 \leq T_s \leq T \leq T_\varepsilon \leq [n/2]$ are T's relative relational degrees.

Let $\eta(T^*) = \max\{\eta(T) | T = T_s, T_{s+1}, \ldots, T_\varepsilon\}$ be the change point of the time series $X = \left( x_{(1)}, x_{(2)}, \ldots, x_{(n)} \right)$, which is related to the maximum $T^*$ value of T's overall relational grade.

Note that, if the change point appears in the latter part of the time series $S = \left( s_{(1)}, s_{(2)}, \ldots, s_{(n)} \right)$, we change to $x_{(k)} = s_{(n-k+1)}, k = 1, 2, \ldots, n$; generally, we assume that $T_s \geq 5$.

### 3.3. GM (1, 1) Forecasting Model

The grey system theory takes a small sample of an uncertain qualitative system with some known information and some unknown information as the research object, and realizes the correct description and effectively monitors the system behavior. The grey forecasting GM (1, 1) model represents the core of the grey system theory. It quantifies the concept of system information sampling, conceptualizes the quantitative model, and finally optimizes the model to predict some unknown data. The GM (1, 1) model has no specific requirements for sample size, and it can be used to research the future time distributions for specific time intervals [55]. Li et al. consider that the GM (1, 1) model is usually used to predict samples with a small amount of data [56]. In our paper, there are only 22 data columns for each industry. For this reason, after considering other methods for uncertain qualitative systems [57,58], we decided to adopt the GM (1, 1) model. The software environment in which we use this approach is the Data Processing System.

Different prediction models can be used in researching energy consumption, such as the autoregressive integrated moving average (ARIMA) model [59], the stacking multi-learning ensemble model [60], and others [61]. However, we consider that the GM (1, 1) model has advantages in terms of calculation quantity and stronger operability. When these models were applied to actual data, scholars found that the GM (1, 1) model's prediction result was more conservative than the ARIMA model's result [59]. Moreover, owing to the different modeling principles and data usage, the signs of the residuals of the GM (1, 1) model and other models differ [59,60]. In general, it is undeniable that certain errors will remain, regardless of what prediction method is used.

Assume that there is a data series $X^{(0)} = \left( x^{(0)}(1), x^{(0)}(2), \ldots, x^{(0)}(n) \right)$. Its accumulating generation operational sequence can be shown, as follows:

$$X^{(1)} = \left( x^{(1)}(1), x^{(1)}(2), \ldots, x^{(1)}(n) \right), \ x^{(1)}(k) = \sum_{i=1}^{k} x^{(0)}(i), \quad k = 1, 2, \ldots, n. \tag{13}$$

The sequence $x^{(1)}$ satisfies a one-order linear differential equation [62]:

$$\frac{dx^{(1)}}{dt} + ax^{(1)} = u \tag{14}$$

where a is the development coefficient and u is the grey action quantity.

The least squares method is used to solve parameters a and u. Afterwards, the prediction model is finally obtained [63]:

$$\hat{X}^{(1)}(k+1) = \left( X^{(0)}(1) - \frac{u}{a} \right) e^{-ak} + \frac{u}{a} \tag{15}$$

We can process the model test as follows.

The original sequence is $X^{(0)}$ and the residual error sequence is $\varepsilon^{(0)}$. $\bar{x} = \frac{1}{n} \sum_{k=1}^{n} x^{(0)}(k)$ is the mean value of $X^{(0)}$ and $s_1^2 = \frac{1}{n} \sum_{k=1}^{n} \left( x^{(0)}(k) - \bar{x} \right)^2$ is the variance of $X^{(0)}$. In the same way, we can calculate the $\bar{\varepsilon}$ and $s_2^2$, which are the mean value and the variance of $\varepsilon^{(0)}$, respectively [63]. We can use $c = \frac{s_2}{s_1}$ to evaluate the model, which can be referred to as the accuracy check level. If $c < 0.35$, the model can be evaluated as "very good"; if $c < 0.50$, the model can be evaluated as "good"; if $c < 0.65$, the model can be evaluated as "poor"; if $c \geq 0.65$, the model can be evaluated as "bad" [63]. Generally speaking, if $c < 0.50$, the model passes the accuracy check level test.

The algorithm flowchart in this paper is shown in Figure 1, which illustrates the three methods that we use. [55,64]. The conclusions drawn arise come from three aspects of the analysis: the analysis of the energy consumption indexes and change points, the characteristics of annual growth rate and the analysis of the forecast lines.

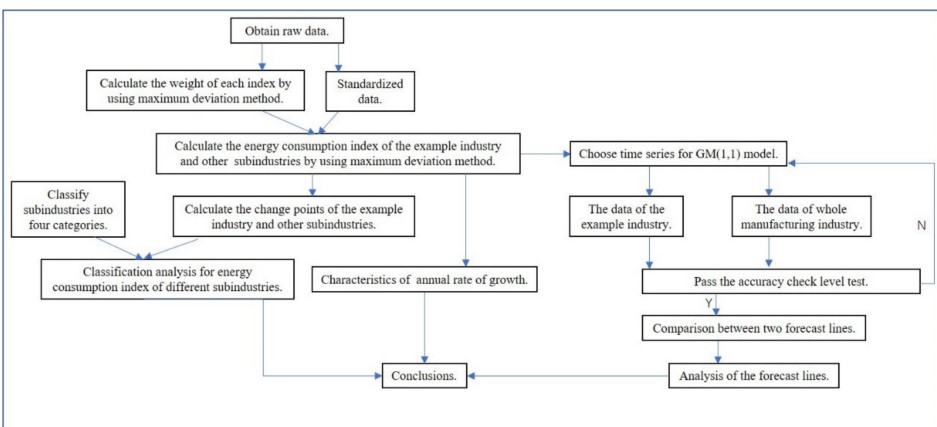

**Figure 1.** Flowchart indicating the methods and structure of the paper.

## 4. Data and Empirical Process

In this section, we explain the source of the data used in this paper, the reasons for choosing the data for these years, how to calculate the energy consumption index, and the change points of a specific industry, using the method described above, and how to analyze the calculation results.

*4.1. Data Sources and Energy Consumption Indexes*

The data in this paper are mainly obtained from the China Statistical Yearbook [65], the China Energy Statistics Yearbook [66], and the China Industrial Statistics Yearbook [67]. China's manufacturing industry categories were basically stable after 1994; therefore, the data used in this paper commence in 1994.

Based on scheme set $A = \{1994, 1995......2015\}$, a total of 22 decision schemes exist, that is, $n = 22$. The index set $G = \{G_1, G_2......G_{10}\}$, that is, $m = 10$, where $G_1$ denotes the energy consumption per unit of output, $G_2$ is coal consumption, $G_3$ is coke consumption, $G_4$ is crude oil consumption, $G_5$ is fuel consumption, $G_6$ is kerosene consumption, $G_7$ is diesel consumption, $G_8$ is fuel oil consumption, $G_9$ is gas consumption, and $G_{10}$ is electric power consumption, all measured on a per unit of output basis. As these indexes are all cost indexes, we can construct the standardization evaluation matrix according to Equation (1) and obtain the weighted vector based on the equations described earlier. Taking the subindustry manufacturing of smelting and pressing of nonferrous metals as an example, for the 10 indexes of consumption per output, we obtain the following weight distribution:

$$\text{w} = (0.1106,\ 0.0819, 0.1073, 0.0876, 0.1027, 0.1082, 0.1027, 0.0820, 0.1001, 0.1169)^T \qquad (16)$$

Based on the weights, the multi-index comprehensive evaluation value $D_i(w)$ can be calculated. For simplicity, we use a series of code names to denote the subindustries of the manufacturing industry, as shown in Table 1.

**Table 1.** Code names and subindustries.

| Code Names | Subindustries |
| --- | --- |
| H1 | manufacturing of agri-food products |
| H2 | manufacturing of foods |
| H3 | manufacturing of beverages |
| H4 | manufacturing of tobacco |
| H5 | manufacturing of textiles |
| H6 | manufacturing of clothing, apparel, and caps |
| H7 | manufacturing of leather, fur, feathers and related products |
| H8 | manufacturing of lumber processing and bamboo, vine, palm, and grass products |
| H9 | manufacturing of furniture |
| H10 | manufacturing of paper and paper products |
| H11 | manufacturing of printing and recording media |
| H12 | manufacturing of articles for culture and educational sport activities |
| H13 | manufacturing for processing of petroleum and nuclear fuel |
| H14 | manufacturing of raw chemical materials and chemical products |
| H15 | manufacturing of medicines |
| H16 | manufacturing of chemical fibers |
| H17 | manufacturing of rubber and plastics |
| H18 | manufacturing of nonmineral product |
| H19 | manufacturing of smelting and pressing of ferrous metals |
| H20 | manufacturing of smelting and pressing of nonferrous metals |
| H21 | manufacturing of metal products |
| H22 | manufacturing of general purpose machinery |
| H23 | manufacturing of special purpose machinery |
| H24 | manufacturing of transport equipment |
| H25 | manufacturing of electrical machinery and equipment |
| H26 | manufacturing of communication equipment, computers and other electronic equipment |
| H27 | manufacturing of measuring instruments and machinery |

It should be noted that if H1 is mentioned below, it means manufacturing of agri-food products. In terms of other code names, we can also get the corresponding industry in Table 1.

The energy consumption indexes of the main divisions of the manufacturing industry are shown in Table 2a, Table 2b and Table 2c.

**Table 2a.** Energy consumption indexes of the subindustries shown in a, b, c.

| Year | H1 | H2 | H3 | H4 | H5 | H6 | H7 | H8 | H9 |
|------|------|------|------|------|------|------|------|------|------|
| 1994 | 0.8203 | 0.7515 | 0.7041 | 0.4801 | 0.6417 | 0.6051 | 0.5502 | 0.6241 | 0.6272 |
| 1995 | 0.8441 | 0.7579 | 0.7315 | 0.6185 | 0.7087 | 0.5627 | 0.5847 | 0.6088 | 0.4874 |
| 1996 | 0.8347 | 0.5644 | 0.4983 | 0.6068 | 0.6678 | 0.6120 | 0.2689 | 0.4924 | 0.4988 |
| 1997 | 0.6549 | 0.5619 | 0.3805 | 0.4753 | 0.5851 | 0.6059 | 0.2476 | 0.3518 | 0.5431 |
| 1998 | 0.7017 | 0.5570 | 0.3804 | 0.5695 | 0.7837 | 0.5813 | 0.3667 | 0.4712 | 0.4552 |
| 1999 | 0.5725 | 0.5784 | 0.4013 | 0.6663 | 0.6412 | 0.6417 | 0.4132 | 0.3515 | 0.3558 |
| 2000 | 0.5419 | 0.4831 | 0.3717 | 0.6180 | 0.5482 | 0.5470 | 0.3164 | 0.2915 | 0.2831 |
| 2001 | 0.4574 | 0.4162 | 0.4472 | 0.5075 | 0.4973 | 0.4944 | 0.2693 | 0.2730 | 0.2570 |
| 2002 | 0.3793 | 0.3611 | 0.3341 | 0.3971 | 0.4553 | 0.4684 | 0.2171 | 0.2312 | 0.2095 |
| 2003 | 0.2725 | 0.2667 | 0.2945 | 0.3632 | 0.3558 | 0.4164 | 0.1940 | 0.2516 | 0.1139 |
| 2004 | 0.1705 | 0.3221 | 0.4421 | 0.2983 | 0.3002 | 0.4415 | 0.3569 | 0.4272 | 0.2350 |
| 2005 | 0.1532 | 0.2539 | 0.3697 | 0.2705 | 0.2165 | 0.3627 | 0.2998 | 0.4301 | 0.1970 |
| 2006 | 0.1201 | 0.1988 | 0.2978 | 0.2295 | 0.1998 | 0.2923 | 0.2630 | 0.3381 | 0.1414 |
| 2007 | 0.0846 | 0.1555 | 0.2291 | 0.1866 | 0.1542 | 0.2269 | 0.2262 | 0.2556 | 0.1056 |
| 2008 | 0.0672 | 0.1477 | 0.2105 | 0.1819 | 0.1230 | 0.2174 | 0.2149 | 0.2319 | 0.1614 |
| 2009 | 0.0527 | 0.1156 | 0.1659 | 0.1431 | 0.0897 | 0.1610 | 0.1904 | 0.2549 | 0.1324 |
| 2010 | 0.0405 | 0.0979 | 0.1237 | 0.1221 | 0.0558 | 0.1444 | 0.1054 | 0.1698 | 0.0887 |
| 2011 | 0.0230 | 0.0834 | 0.1021 | 0.1184 | 0.0281 | 0.1426 | 0.1247 | 0.1402 | 0.0810 |
| 2012 | 0.0130 | 0.0945 | 0.1062 | 0.1421 | 0.0262 | 0.1064 | 0.1685 | 0.1085 | 0.0907 |
| 2013 | 0.0166 | 0.0942 | 0.1111 | 0.1245 | 0.0213 | 0.1064 | 0.1693 | 0.0961 | 0.0848 |
| 2014 | 0.0130 | 0.0911 | 0.1212 | 0.1097 | 0.0080 | 0.1037 | 0.1339 | 0.1158 | 0.1017 |
| 2015 | 0.0586 | 0.0931 | 0.1245 | 0.0931 | 0.0138 | 0.0887 | 0.1346 | 0.1048 | 0.1289 |

**Table 2b.** Energy consumption indexes of the subindustries shown in a, b, c.

| Year | H10 | H11 | H12 | H13 | H14 | H15 | H16 | H17 | H18 |
|------|------|------|------|------|------|------|------|------|------|
| 1994 | 0.7987 | 0.6786 | 0.4481 | 0.7079 | 0.9945 | 0.8732 | 0.8509 | 0.8122 | 0.7364 |
| 1995 | 0.6741 | 0.6128 | 0.3936 | 0.7674 | 0.7659 | 0.8035 | 0.6653 | 0.7020 | 0.7975 |
| 1996 | 0.5809 | 0.5066 | 0.5147 | 0.5730 | 0.8223 | 0.7757 | 0.7002 | 0.6386 | 0.7203 |
| 1997 | 0.5208 | 0.4227 | 0.4294 | 0.6605 | 0.6910 | 0.5108 | 0.6801 | 0.5589 | 0.6295 |
| 1998 | 0.5732 | 0.5353 | 0.5833 | 0.7219 | 0.6665 | 0.3997 | 0.8307 | 0.7207 | 0.7579 |
| 1999 | 0.5622 | 0.6154 | 0.6132 | 0.6925 | 0.5992 | 0.4561 | 0.6377 | 0.5871 | 0.6924 |
| 2000 | 0.4836 | 0.5688 | 0.5453 | 0.4055 | 0.5246 | 0.3807 | 0.5346 | 0.5352 | 0.6501 |
| 2001 | 0.4217 | 0.5085 | 0.5600 | 0.4423 | 0.4680 | 0.3424 | 0.6657 | 0.4838 | 0.5681 |
| 2002 | 0.3849 | 0.4253 | 0.5059 | 0.4456 | 0.4366 | 0.3117 | 0.5985 | 0.3840 | 0.5065 |
| 2003 | 0.3111 | 0.3589 | 0.3435 | 0.3767 | 0.3548 | 0.2446 | 0.4805 | 0.3330 | 0.4059 |
| 2004 | 0.2254 | 0.2984 | 0.3410 | 0.2537 | 0.2359 | 0.2736 | 0.2653 | 0.3084 | 0.3709 |
| 2005 | 0.1861 | 0.2262 | 0.2927 | 0.1419 | 0.1979 | 0.2227 | 0.1998 | 0.2832 | 0.3361 |
| 2006 | 0.1545 | 0.1990 | 0.2419 | 0.1103 | 0.1584 | 0.1830 | 0.1584 | 0.2212 | 0.2607 |
| 2007 | 0.1130 | 0.1654 | 0.2285 | 0.0924 | 0.1182 | 0.1456 | 0.1244 | 0.1595 | 0.1935 |
| 2008 | 0.1025 | 0.1906 | 0.2046 | 0.0550 | 0.0858 | 0.1475 | 0.1010 | 0.1391 | 0.1593 |
| 2009 | 0.0802 | 0.1675 | 0.1911 | 0.0801 | 0.0692 | 0.1165 | 0.0711 | 0.0918 | 0.1167 |
| 2010 | 0.0503 | 0.1610 | 0.1846 | 0.0449 | 0.0398 | 0.1072 | 0.0341 | 0.0702 | 0.0671 |
| 2011 | 0.0395 | 0.1264 | 0.1520 | 0.0279 | 0.0230 | 0.0752 | 0.0136 | 0.0303 | 0.1568 |
| 2012 | 0.0489 | 0.1080 | 0.1107 | 0.0326 | 0.0104 | 0.0768 | 0.0085 | 0.0264 | 0.0655 |
| 2013 | 0.0652 | 0.1102 | 0.1110 | 0.0505 | 0.0156 | 0.0707 | 0.0146 | 0.0198 | 0.0538 |
| 2014 | 0.0590 | 0.1224 | 0.1196 | 0.0535 | 0.0145 | 0.0571 | 0.0132 | 0.0159 | 0.0475 |
| 2015 | 0.0840 | 0.1152 | 0.1046 | 0.0925 | 0.0144 | 0.0514 | 0.0154 | 0.0117 | 0.0341 |

**Table 2c.** Energy consumption indexes of the subindustries shown in a, b, c.

| Year | H19 | H20 | H21 | H22 | H23 | H24 | H25 | H26 | H27 |
|------|------|------|------|------|------|------|------|------|------|
| 1994 | 0.8385 | 0.9068 | 0.8714 | 0.9632 | 0.9123 | 0.8489 | 0.9674 | 0.9120 | 0.6055 |
| 1995 | 0.8062 | 0.5878 | 0.8077 | 0.7338 | 0.7748 | 0.8070 | 0.7253 | 0.5829 | 0.6857 |
| 1996 | 0.8073 | 0.6746 | 0.7472 | 0.7336 | 0.6999 | 0.6498 | 0.7115 | 0.6654 | 0.4483 |
| 1997 | 0.7483 | 0.7060 | 0.6122 | 0.6577 | 0.5083 | 0.6053 | 0.6643 | 0.5455 | 0.3227 |
| 1998 | 0.7796 | 0.6094 | 0.5795 | 0.5836 | 0.6113 | 0.6824 | 0.5566 | 0.5552 | 0.4261 |
| 1999 | 0.6974 | 0.5001 | 0.5906 | 0.5183 | 0.5715 | 0.5788 | 0.4537 | 0.4711 | 0.4659 |
| 2000 | 0.5764 | 0.4185 | 0.5198 | 0.4333 | 0.4964 | 0.5074 | 0.3459 | 0.3479 | 0.3432 |
| 2001 | 0.4984 | 0.3925 | 0.5012 | 0.3606 | 0.4314 | 0.4529 | 0.2947 | 0.3170 | 0.3592 |
| 2002 | 0.4803 | 0.3897 | 0.4894 | 0.3230 | 0.3579 | 0.3146 | 0.3039 | 0.3057 | 0.3393 |
| 2003 | 0.3152 | 0.3433 | 0.4468 | 0.2970 | 0.2924 | 0.2338 | 0.2894 | 0.2303 | 0.2752 |
| 2004 | 0.1703 | 0.2653 | 0.3420 | 0.2387 | 0.2470 | 0.2827 | 0.2323 | 0.1657 | 0.1722 |
| 2005 | 0.1631 | 0.2204 | 0.2710 | 0.1828 | 0.2045 | 0.2845 | 0.1933 | 0.1446 | 0.1467 |
| 2006 | 0.1507 | 0.1178 | 0.2128 | 0.1463 | 0.1466 | 0.2142 | 0.1389 | 0.1252 | 0.1216 |
| 2007 | 0.1017 | 0.0784 | 0.1460 | 0.1030 | 0.1042 | 0.1561 | 0.0995 | 0.1108 | 0.1065 |
| 2008 | 0.0528 | 0.0683 | 0.1272 | 0.0769 | 0.0785 | 0.1935 | 0.0912 | 0.1107 | 0.1468 |
| 2009 | 0.0757 | 0.0757 | 0.1182 | 0.0656 | 0.0556 | 0.1410 | 0.0778 | 0.1019 | 0.1818 |
| 2010 | 0.0436 | 0.0530 | 0.1041 | 0.0455 | 0.0449 | 0.0996 | 0.0614 | 0.0799 | 0.1818 |
| 2011 | 0.0267 | 0.0435 | 0.0684 | 0.0390 | 0.0224 | 0.1116 | 0.0365 | 0.0354 | 0.1142 |
| 2012 | 0.0209 | 0.0850 | 0.0598 | 0.0317 | 0.0099 | 0.1359 | 0.0285 | 0.0253 | 0.1425 |
| 2013 | 0.0324 | 0.0930 | 0.0923 | 0.0180 | 0.0144 | 0.1323 | 0.0257 | 0.0330 | 0.1462 |
| 2014 | 0.0440 | 0.1072 | 0.0908 | 0.0113 | 0.0134 | 0.1184 | 0.0104 | 0.0323 | 0.1295 |
| 2015 | 0.0653 | 0.1181 | 0.0951 | 0.0114 | 0.0029 | 0.0873 | 0.0033 | 0.0328 | 0.1369 |

From the data results, we can see that the energy consumption index of each industry differs from that of other industries in the starting year (1994), with some being as high as 0.9, and other less than 0.5. However, the common feature is that the energy consumption indexes fall as the years progress, that is, later years have lower indexes. After 2011, the energy consumption indexes of several industries rebound, including in the manufacturing of paper and paper products (H10) and the manufacturing of smelting and pressing of nonferrous metals (H20) industries. The energy consumption indexes for most industries remained below 0.1 in 2015. Subindustries H1, H14, H15, H16, H17, H19, H21, H23, H24, H25, and H26 are among the better performing industries because their energy consumption indexes generally exceeded 0.8 in 1994 but had dropped to less than 0.1 by 2015, which was a significant and rapid decline. These industries include both traditional energy-intensive industries and high-tech industries. The underperforming sectors are H7, H8, H9, H11, H12, and H27. The energy consumption indexes of these industries were between 0.4 and 0.7 in 1994, and they remained above 0.1 in 2015, which is a relatively small decline. The energy-intensive industries include but are not limited to H13, H17, H19, and H20. To analyze the specific energy consumption index of each industry, we will convert these data into line graphs that will be analyzed in detail in the next section, along with our analysis of the distribution of change points.

*4.2. Selection of Change Points*

By using the data in Table 2 and grey relational analysis, the change points in the data series for the energy consumption indexes can be determined. If we simply observe, rather than using mathematical methods, we may obtain the right "inflection point" on a data column. However, the purpose of using the change point search method is to validate our intuitive guesses in a mathematical way, so that we can segment the data columns more logically. Again, we take the subindustry manufacturing of smelting and pressing of nonferrous metals (H19) as an example. Looking at the energy consumption index for this industry, we would guess, based on the approximate growth rate, that the first change point would be in 1999 or 2000 and the second change point would be in 2006 or 2007. However, we do not have clear evidence to prove this. Therefore, we used the change point search of grey relational

analysis as a mathematical method to find the change points on the data columns. The numerical results of the energy consumption index for this subindustry are shown in Tables 3 and 4 below.

**Table 3.** Numeric results of the relative variety ratio of the relational degree and variety degree for the series (part 1).

| T | Dissected Time Series | $\eta(T)$ | T* | Change Point |
|---|---|---|---|---|
| 5 | 1999–2015/1994–1998 | 4.4201 | | |
| 6 | 2000–2015/1994–1999 | 1.3723 | | |
| 7 | 2001–2015/1994–2000 | 9.1666 | 7 | 2000 |
| 8 | 2002–2015/1994–2001 | 5.4313 | | |
| 9 | 2003–2015/1994–2002 | 4.9794 | | |
| 10 | 2004–2014/1985–2003 | 0.0000 | | |

**Table 4.** Numeric results of the relative variety ratio of the relational degree and variety degree for the series (part 2).

| T | Dissected Time Series | H(T) | T* | Change Point |
|---|---|---|---|---|
| 5 | 1994–2010/2011–2015 | 4.7640 | | |
| 6 | 1994–2009/2010–2015 | 4.9641 | | |
| 7 | 1994–2008/2009–2015 | 3.5634 | 9 | 2007 |
| 8 | 1994–2007/2008–2015 | 4.5347 | | |
| 9 | 1994–2006/2007–2015 | 18.7681 | | |
| 10 | 1994–2005/2006–2015 | 0.0000 | | |

The maximum point in Table 3 is the seventh point, so the change of the time series occurs in 2000. The other change point is shown in Table 4, and the maximum point is the ninth point, that is, the change point occurs in 2007. Thus, the time series can be divided into three parts: 1994–2000, 2001–2007, and 2008–2015.

We present the data on the subindustry manufacturing of smelting and pressing of nonferrous metals from Table 1 in bar column format in Figure 1, and we plot the annual growth rate with a red line. The energy consumption index of this subindustry shows a general downward trend from 1994 to 2015, although the index remains at a high level. The trend experiences fluctuations before 2000, but a continuous downward trend is shown after 2001, with the index dropping from 0.4 to 0.1. Since 2008, the index has remained below 0.1, although there is a slight rebound trend in 2011. However, by 2015, the latest year for which data are available, the index does not rise to 0.2, but it remains at a low level. Therefore, it is reasonable to select 2000 and 2007 as change points for the energy consumption index of this manufacturing industry (See Figure 2).

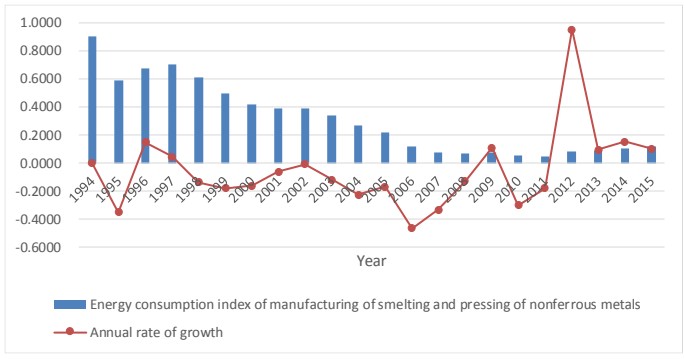

**Figure 2.** Trend of the energy consumption index and the annual rate of growth for the subindustry manufacturing of smelting and pressing of nonferrous metals.

## 5. Analysis of the Subindustries of the Manufacturing Industry

In this section, we will first summarize the change points of 27 subindustries, and then classify these industries into four categories and analyze each of them in detail. These categories are the consumer products industry, the raw materials industry, the equipment industry and other subindustries of the manufacturing industry.

### 5.1. Summary of Change Points

The change points of the energy consumption indexes of different industry subdivisions can be determined, as shown in Table 5.

**Table 5.** Change points of the energy consumption index of 27 subindustries.

| Subindustries | Change Point 1 | Change Point 2 |
| --- | --- | --- |
| H1 | 1999 | 2010 |
| H2 | 2002 | 2007 |
| H3 | 2001 | 2008 |
| H4 | 2002 | 2007 |
| H5 | 1998 | 2008 |
| H6 | 1998 | 2008 |
| H7 | 2000 | 2007 |
| H8 | 2002 | 2007 |
| H9 | 2002 | 2007 |
| H10 | 1999 | 2008 |
| H11 | 2000 | 2008 |
| H12 | 2002 | 2007 |
| H13 | 1998 | 2010 |
| H14 | 2000 | 2007 |
| H15 | 1999 | 2008 |
| H16 | 2001 | 2007 |
| H17 | 2001 | 2008 |
| H18 | 2001 | 2007 |
| H19 | 1998 | 2011 |
| H20 | 2000 | 2007 |
| H21 | 1999 | 2008 |
| H22 | 1998 | 2008 |
| H23 | 1998 | 2007 |
| H24 | 1998 | 2008 |
| H25 | 2002 | 2007 |
| H26 | 2001 | 2009 |
| H27 | 1999 | 2010 |

We find that the change points of the time series are mainly concentrated in 2000 and 2008. The year 2000 marks the point at which China sought to transform the extensive model of its manufacturing industry, and when it began to prepare actively to enter the World Trade Organization. Around this time, most companies responded to the national call for mandatory or voluntary efforts to conserve energy and reduce emissions. Thus, the energy consumption index drops rapidly at this time. The second change point is around 2008, when the economic crisis was sweeping the world. As a large country opening to the outside world, it was inevitable that China was also affected and many of its companies faced crises and were forced to reduce production. Studies have shown that the global financial crisis led to China's energy consumption falling by nearly 10% [68]. As the manufacturing industry is the core of the industrial economy, a further drop of the energy consumption indexes in the subindustries of the manufacturing sector is understandable.

### 5.2. Analysis of the Change Points of the Consumer Products Industry

The main products of 14 subindustries can be categorized as consumer products. This includes the subindustries agri-food products, manufacturing of foods, and manufacturing of beverages, and all subindustries listed in the Figures 3 and 4. Based on the trends in the energy consumption indexes of these industries, they can be divided into two categories. The specific trends are shown in Figures 3 and 4.

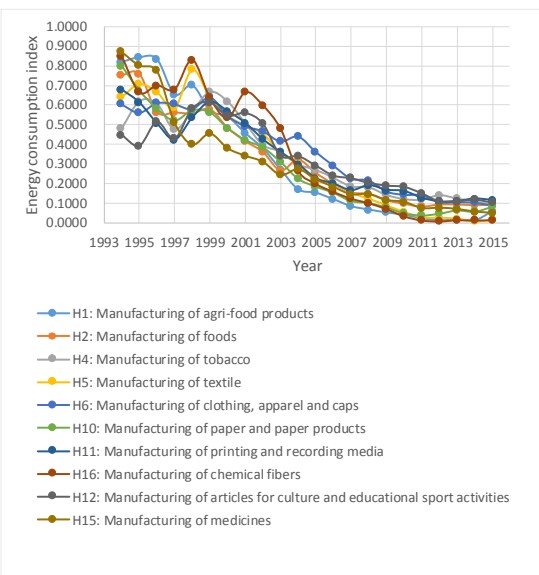

**Figure 3.** Trends in the energy consumption index of the consumer products industry (part 1).

The change points of the energy consumption indexes in the manufacturing of textiles, apparel and caps, paper and paper products, and medicines are detected in 1999 and 2008. In the manufacturing of foods, tobacco, and articles for culture, educational, and sports activities, change points are determined in 2002 and 2007. In the manufacturing of chemical fibers, the change points occur in 2001 and 2007. In the manufacturing of printing and recording media, the change points occur in 2000 and 2008. In the manufacturing of agri-food products, the change points are detected in 1999 and 2010. Figure 3 shows the trends in the energy consumption indexes for these 10 industries. The change points divide the cycle of the energy consumption indexes into three phases: in the first phase, the indexes fluctuate up and down, without a notable downward trend; in the second phase, the downward trend is relatively marked; and, in the third phase, there is a downward trend, but the amplitude is smaller. By 2015, the change points have resulted in the indexes of these industries being concentrated between 0 and 0.1.

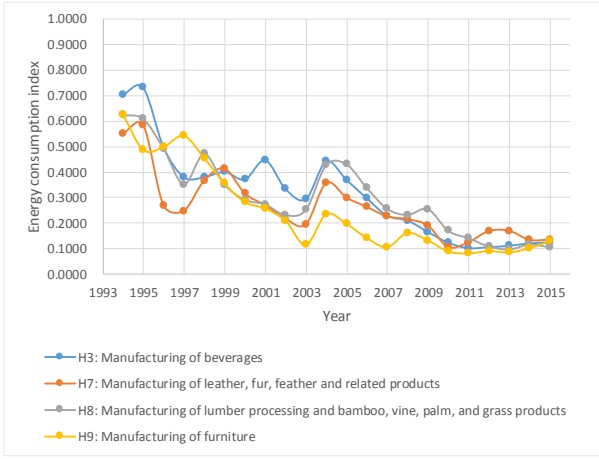

**Figure 4.** Trends in the energy consumption index of the consumer products industry (part 2).

Change points in the manufacturing of furniture and lumber processing and bamboo, vine, palm, and grass products are detected in 2002 and 2007. They occur in 2001 and 2008 in the manufacturing of beverages and in 2000 and 2007 in the manufacturing of leather, fur, feather, and related products. Figure 4 shows that the change points divide the cycle of the energy consumption index into three phases: in the first phase, the trend of the energy consumption indexes fluctuates, but there is a significant downward trend; in the second phase, the indexes fluctuate between 0.1 and 0.5; in the third phase, the indexes of these four industries drop by approximately 0.2 and stabilize at around 0.1 in 2015.

The characteristics of the industries in Figures 3 and 4 differ slightly, although, because they all belong to the consumer goods industry category, similarities can be perceived. However, the decline in the energy consumption indexes of the industries in Figure 3 is more obvious, whereas the indexes in Figure 4 experienced more fluctuations. These differences may arise from the production and sales characteristics of the industries in question, each of which respond slightly differently to energy policies, resulting in slight differences in the trends and characteristics of their energy consumption indexes.

Most industries in the consumer goods industry are competitive, and they face problems, such as decentralization, severe homogenization, uneven technical levels and a scarcity of big brands. From the perspective of energy consumption, the subindustry manufacturing of textiles and chemical fibers is a significant consumer of electricity. After 1998, and especially during the period of the Tenth Five-Year Plan (2001–2005), the energy consumption per unit of output of the light industries, such as food, medicine, textiles, and papermaking, declined significantly. The changes in the energy consumption indexes of these industries and the creation of change points are related to industry policies. The key words for these industries in the *Tenth Five-Year Plan* were upscale, differentiated, and deep processing. National policies encourage the development of high-end products and the elimination of high-energy-consuming products that lack competitiveness. In addition, more consumer goods and industrial products entered the international market after China formally joined the World Trade Organization in 2001, which meant that the technologies of the related industries became aligned with international standards. Furthermore, the Chinese government has organized special projects to tackle high-energy-consumption problems, for example, deep processing technology for the food industry, projects for innovative drugs and the modernization of traditional Chinese medicine, clean production technology for the leather industry, audit evaluation projects to encourage clean production, and energy audits of printing and dyeing companies. These government-led initiatives promoted the initial effective development of deep processing in the consumer goods industry, directly resulting in the downward trend in its energy consumption index. In 2009, the *Light Industry Adjustment and Revitalization Plan* was published and the energy consumption policies of the consumer goods industry were further advanced.

The *Key Energy Conservation Technology Promotion Catalogue (First Batch)* encouraged software and hardware construction for a statistical monitoring system for energy saving and emission reductions in the food, paper, and other consumer products industries, and established an exit mechanism for these industries, in line with plans to eliminate a group of industries with outdated, high-energy-use production facilities within three years. For example, the paper industry phased out grass pulp production equipment with an annual output of 34,000 tons or less. Under the guidance of various five-year plans in the early 21st century, an intensive and energy-saving consumer goods industry is taking shape. The consumer products industry has carried out many resource integration activities and profits are gradually being absorbed by large enterprises with relatively advanced technologies in various fields. The result is that the energy consumption indexes of these industries maintain a continuous downward trend.

*5.3. Analysis of the Change Points of the Raw Materials Industry*

The raw materials industry includes the manufacturing of rubber and plastics, nonmineral products, and four other industries, as shown in Figure 5, which indicates specific trends. In general, the raw material industries tend to be more energy intensive.

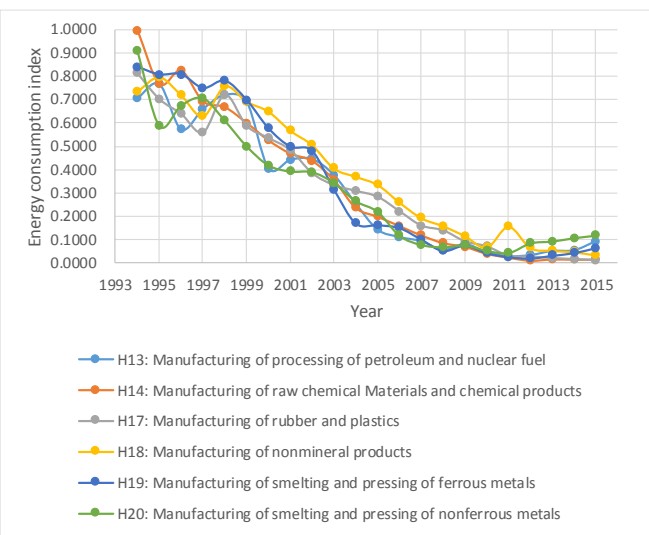

**Figure 5.** Trends in the energy consumption indexes of the raw materials industry.

In the manufacturing of smelting and pressing of nonferrous metals and the manufacturing of raw chemical materials and chemical products, the change points occur in 2000 and 2007. In the manufacturing of smelting and pressing of ferrous metals, the change points are detected in 1998 and 2011. In the manufacturing of processing of petroleum and nuclear fuel, the change points occur in 1998 and 2010. In the manufacturing of rubber and plastics, the change points are determined in 2001 and 2008. In the manufacturing of nonmineral products, the change points occur in 2001 and 2007. Figure 5 shows the trends in the energy consumption indexes of these six industries. Except for the manufacturing of nonmineral products and raw chemical materials and chemical products, the indexes of the other four industries have a rising trend in the third stage.

High energy consumption has always been a feature of the raw materials industry. Due to the advancement of industrialization and urbanization, society's consumption of the products of these industries has been enormous, and blind expansion has resulted in overcapacity. At the beginning of the 21st century, the government successively issued policies for structural adjustment and energy saving for high-energy-consuming industries, including the *Notice on Accelerating the Adjustment of the Industry Structure with Overcapacity (2006)* and the *Urgent Notice on Accelerating the Adjustment of Industrial Structure and Restraining the Blind Expansion of High Energy-consuming Industries Again (2007)*. In 1999, the China Iron and Steel Association was established with the goal of establishing an industry self-discipline mechanism. The key words for the development of these industries in the *10th* and *11th Five-Year Plans* are refinement, deep processing, new types and high efficiency. During the period of the *12th Five-Year Plan*, these industries remained the key focus for rectification, with national policies continuing to support adjustment of their industrial layout. The raw material industry has been committed to retrofitting measures, such as eliminating low-end steel production capacity and inhibiting excessive growth of the flat glass production capacity. In addition to tax incentives and subsidies, the industry has used high-performance, environmentally friendly materials to reduce its energy consumption.

As of 2015, the energy consumption index of these high-energy-consuming industries is basically stable, at below 0.1, whereas the annual output value of their industrial products is climbing, which is reflective of the results that were achieved by the industry in reducing consumption and conserving energy, with the support of government policies.

*5.4. Analysis of the Change Points of the Equipment Industry*

The three subindustries that manufacture general purpose machinery, special purpose machinery, and transport equipment belong to the equipment industry. The trends in the energy consumption indexes of these three industries are shown in Figure 6.

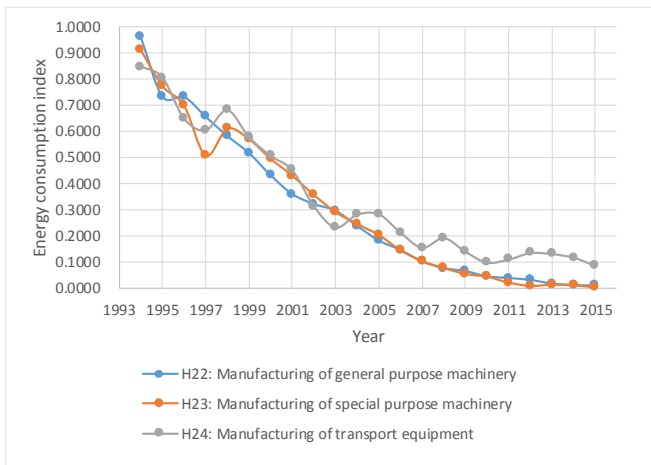

**Figure 6.** Trends in the energy consumption indexes of the equipment industry.

In the manufacturing of general purpose machinery and the manufacturing of transport equipment, change points occur in 1998 and 2008. For the manufacturing of special purpose machinery, they occur in 1998 and 2007. In 2015, the energy consumption index for the manufacturing of transport equipment remained at around 0.1, whereas the indexes of the other two industries were close to zero.

Since China's reform and opening, its equipment industry has faced difficulties that are associated with weak foundations and a lack of core technologies. At the end of the 20th century, China's equipment manufacturing system greatly improved and its application areas greatly expanded. The *10th Five-Year Plan* proposed to revitalize this industry, focusing on the development of computer numerical control machine tools and basic components; support for new types of high-efficiency power generation equipment, such as nuclear power units; and active development of high-efficiency, energy-saving, low-emission vehicle engines, and hybrid power systems. During this period, the energy consumption per unit of output value of these three subindustries has declined to a great extent and the indexes have fallen by more than 0.4 from 1998 to 2005.

At the start of the *11th Five-Year Plan*, the equipment industry still faced many development issues. To further promote the development of the high-end equipment industry, the State Council issued several opinions on Accelerating the Rejuvenation of the Equipment Manufacturing Industry (2006) and the People's Assembly passed the Industrial Transformation and Upgrade Plan (2011–2015), proposing to actively cultivate and develop intelligent manufacturing and new energy vehicles. During the *12th Five-Year Plan* period, the investment in scientific research for the equipment industry greatly increased, which resulted in greater scientific research capacity and technological progress. Based on the capacity increase, these policies effectively reduced the energy consumption per unit of output value for these subindustries.

*5.5. Analysis of the Change Points of Other Subindustries of the Manufacturing Industry*

Four subindustries classified as "other" subindustries, are analyzed in this section. In the manufacturing of metal products, the change points are in 1999 and 2008. They occur in 2001 and 2009 in the manufacturing of communication equipment, computers, and other electronic equipment. Change points are detected in 2002 and 2007 in the manufacturing of electrical machinery and equipment, and in 1999 and 2010 in the manufacturing of measuring instruments and machinery. Figure 7 shows the trends in the energy consumption indexes of these four subindustries. In the third

phase, the indexes are stable at around 0.1. However, with the exception of the electrical manufacturing of machinery and equipment, the energy consumption indexes rebounded slightly in 2015.

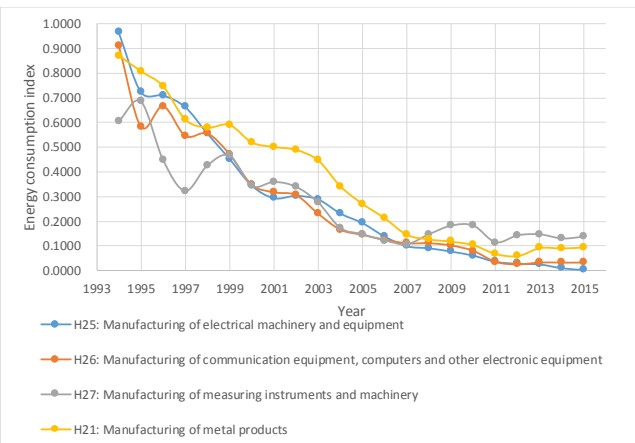

**Figure 7.** Trends in the energy consumption indexes of "other" subindustries of the manufacturing industry.

Manufacturing of electrical machinery and equipment and manufacturing of communication equipment, computers and other electronic equipment are the fastest growing industries, in the context of the arrival of the information age. In the *10th Five-Year Plan*, the promotion of high-tech industrialization of digital electronic products, new display devices, optoelectronic materials, and devices was cited. These two subindustries have achieved good results during this period and China has become an important manufacturing base for electronic information products worldwide. During the period of the *11th Five-Year Plan*, these industries further optimized their industrial structure, extended their industrial chains, and paid attention to the construction of major projects, such as integrated circuits and industrial bases. Relevant policies during this period included the Measures for the Control of Pollution Caused by Electronic Information Products (2007) and the Procedures for the Development of Catalogue for the Control of Pollution in Electronic Information Products (2008). These policies have had an impact on energy-saving measures in these two industries after 2009, causing their energy consumption indexes to fall below 0.1.

As the degree of marketization in China has deepened, ownership reforms in the manufacturing of measuring instruments and machinery have integrated industry resources, promoting intensive development, eliminating high-energy-consuming, low-end enterprises, and optimizing the industrial structure. Based on the increasingly wide range of service targets for the manufacturing of measuring instruments and machinery, the government issued the Measures for the Management of High-tech Enterprise Recognition (2008), putting forward key support measures for the development of high-performance and intelligent instrumentation. In 2011, the government invested 1.3 billion yuan in the research and development of scientific instruments, signaling the country's increasing focus on scientific instruments. In 2013, the Accelerated Action Plan for the Development of Sensors and Intelligent Instrumentation Industry was published to plan goals, ideas, and actions for the intelligent instrumentation industry. After 2011, the energy consumption index of this industry stabilized between 0.1 and 0.2.

*5.6. Analysis of Annual Rates of Growth*

Based on the trend graphs of the subindustries, change points of the energy consumption indexes of most industries occur around 2000 and 2008. We continue to focus on the manufacturing of smelting and pressing of nonferrous metals subindustry as an example. The trend in the annual rate of growth of this subindustry, as shown in Figure 1, is similar to that of other subindustries in the manufacturing industry. The annual growth rate maintains a steady downward trend in the second stage, as divided by the change points. The reaction speed of different industries is not exactly

the same. For instance, in our example industry, one to three years after the centralized promulgation and implementation of relevant policies, the decline in the growth rate of this industry is more obvious. This indicates that the policies were effective, with a delayed effect in the short term. We obtain similar results for other subindustries. Thus, the results show that the changes in the industries' growth patterns were broadly in line with the timing of national policies, which indicates that the national policies were soundly based and effective [69].

Previous studies [28,32,69] have shown that policies are effective only by matching the characteristics of energy-related indicators with the characteristics of policies issued. This paper starts with such an analysis but adds value to it by focusing on the annual rate of growth of the energy consumption indexes. As a result, we find that policy has a lagged effect in the short term. This is an original contribution by this paper and a supplement to the previous studies.

## 6. Trend Forecast

The grey system theory takes a small sample uncertain qualitative system with some known information and some unknown information as the research object, and realizes the correct description and effectively monitors the system behavior. The grey forecasting GM (1, 1) model represents the core of the grey system theory. It quantifies the concept of system information sampling, conceptualizes the quantitative model, and finally optimizes the model to predict some unknown data.

Based on the grey forecast GM (1, 1) model and the data for the manufacturing industry as a whole from 2001 to 2015, the trends in the energy consumption index for the 10 years after 2015 can be forecast. We also forecast the future trends in the energy consumption index for our example subindustry, the manufacturing of smelting and pressing of nonferrous metals. According to the known energy consumption index, there is a similarity between the whole manufacturing industry and our example subindustry in that the distribution year of the change point is similar. However, the energy consumption index of our subindustry began to rise from 2012, whereas the index for the whole manufacturing industry declined from 2012. The example subindustry is a high energy-consuming industry, and other high energy-consuming industries also have a rebound trend, including the manufacturing of paper and paper products and manufacturing of printing and recording media, manufacturing of smelting, and pressing of ferrous metals. We plan to use this industry as representative and forecast trends for this type of industry. The relevant data for forecasting are shown in Tables 6 and 7.

**Table 6.** Grey model (GM) (1, 1) model's prediction of the energy consumption index of the whole manufacturing industry.

| No. | Observed Value | Fitted Value | Error | % |
|-----|---------------|--------------|-------|---|
| X(2) | 0.5238 | 0.5639 | −0.0401 | −7.65 |
| X(3) | 0.4656 | 0.4407 | 0.0249 | 5.3465 |
| X(4) | 0.3748 | 0.3444 | 0.0304 | 8.1004 |
| X(5) | 0.2726 | 0.2692 | 0.0034 | 1.2499 |
| X(6) | 0.2262 | 0.2104 | 0.0158 | 6.9874 |
| X(7) | 0.1824 | 0.1644 | 0.0180 | 9.8471 |
| X(8) | 0.1314 | 0.1285 | 0.0029 | 2.1976 |
| X(9) | 0.0934 | 0.1004 | −0.0070 | −7.5267 |
| X(10) | 0.0763 | 0.0785 | −0.0022 | −2.8760 |
| X(11) | 0.0462 | 0.0613 | −0.0151 | −32.7210 |
| X(12) | 0.0267 | 0.0479 | −0.0212 | −79.2841 |
| X(13) | 0.0200 | 0.0375 | −0.0175 | −86.9355 |
| X(14) | 0.0204 | 0.0293 | −0.0089 | −43.3559 |
| X(15) | 0.0145 | 0.0229 | −0.0084 | −57.4664 |
| X(16) | 0.0092 | 0.0179 | −0.0087 | −93.4391 |

**Table 7.** GM (1, 1) model's prediction of the energy consumption index for the subindustry manufacturing of smelting and pressing of nonferrous metals.

| No. | Observed Value | Fitted Value | Error | % |
|-----|---------------|--------------|-------|---|
| X(2) | 0.3897 | 0.3713 | 0.0184 | 4.7109 |
| X(3) | 0.3433 | 0.3061 | 0.0372 | 10.8239 |
| X(4) | 0.2653 | 0.2524 | 0.0129 | 4.8702 |
| X(5) | 0.2204 | 0.2081 | 0.0123 | 5.5973 |
| X(6) | 0.1178 | 0.1715 | −0.0537 | −45.5668 |
| X(7) | 0.0784 | 0.1414 | −0.0630 | −80.2599 |
| X(8) | 0.0683 | 0.1166 | −0.0483 | −70.5756 |
| X(9) | 0.0757 | 0.0961 | −0.0204 | −26.9176 |
| X(10) | 0.0530 | 0.0792 | −0.0262 | −49.3934 |
| X(11) | 0.0435 | 0.0653 | −0.0218 | −50.0357 |
| X(12) | 0.0850 | 0.0538 | 0.0312 | 36.6084 |
| X(13) | 0.0930 | 0.0444 | 0.0486 | 52.2117 |
| X(14) | 0.1072 | 0.0366 | 0.0706 | 65.8006 |
| X(15) | 0.1181 | 0.0302 | 0.0879 | 74.3911 |

The predicted energy consumption index for China's manufacturing industry for various years prior to 2025 is obtained, as follows: X(t + 1) = 0.0140, X(t + 2) = 0.0109, X(t + 3) = 0.0085, X(t + 4) = 0.0067, X(t + 5) = 0.0052, X(t + 6) = 0.0041, X(t + 7) = 0.0032, X(t + 8) = 0.0025, X(t + 9) = 0.0020, X(t + 10) = 0.0015. The value of c is 0.0947, so the model of the example industry can be evaluated as "very good". The predicted energy consumption index for the subindustry, manufacturing of smelting and pressing of nonferrous metals, is obtained, as follows: X(t + 1) = 0.0249, X(t + 2) = 0.0205, X(t + 3) = 0.0169, X(t + 4) = 0.0139, X(t + 5) = 0.0115, X(t + 6) = 0.0095, X(t + 7) = 0.0078, X(t + 8) = 0.0064, and X(t + 9) = 0.0053, X(t + 10) = 0.0044. The value of c is 0.3726, so the model of the example industry can be evaluated as "good". The trend chart is shown in Figure 8.

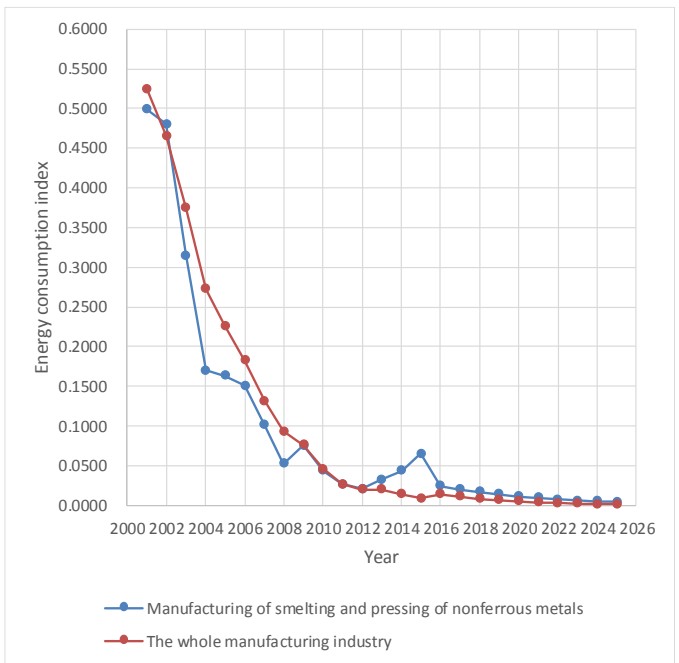

**Figure 8.** Trend forecast of the energy consumption index before 2025.

As shown in Figure 8, although the energy consumption index of the example subindustry has risen in the four years following 2012, the forecast index shows a slight downward trend up to 2025. The energy consumption index for the whole manufacturing industry showed only a slight increase prior to 2016, but then it also declines up to 2025. More precisely, in the prediction stage, the energy

consumption index of the example industry has been declining, whereas the index for the overall manufacturing industry has first risen and then falls. As a result, it is unlikely that the rise in the index of the overall manufacturing industry is caused by industries, such as the example industry.

Traditionally, the energy consumption mode of high-energy-consuming industries requires adjustment after a certain period of time; as such, industries adopt more sustainable development processes. Thus, past studies have generally assumed that enhancing energy efficiency in energy-intensive industries would be more difficult than in other industries [70,71]. In other words, if the energy consumption indexes of high-energy-consuming industries rise, an increase will occur in the overall energy consumption index of the manufacturing industry.

However, the predicted results show the opposite. The forecast results show a positive trend for the energy consumption index of the traditional high energy-consuming industries before 2025. More concretely, the small growth that is forecast for the indexes of the whole manufacturing industry may not be caused by the energy-intensive industries. This is probably because the Chinese government and society have focused on reducing the energy use of these traditional high energy-consuming industries, and thus awareness of the need for energy conservation and emission reductions in these industries has been constantly strengthened. Consequently, in the future, the energy-saving and emission reduction efforts for China's manufacturing industry should be placed on traditional industries other than the high energy-consuming industries.

Based on the *Made in China 2025* policy, China's manufacturing industry is focusing on the integration of information technology and manufacturing, with innovation-driven and green development as its basic principles. Whether the whole manufacturing industry can further reduce the energy consumption index depends on the joint efforts of the enterprises and the Chinese government. Because of the continuous increase in the intensity of environmental regulations and the development of new technologies, various industries in the manufacturing industry have made, and will continue to make, strategic adjustments, and the trend of the energy consumption index in China's manufacturing industry will show a decline in the foreseeable future.

## 7. Conclusions

Based on the above empirical analysis, we obtain the four following key conclusions:

### 7.1. Positive Expectations of the Energy-Intensive Industries

Although there has been a trend of recovery in energy consumption in some energy-intensive industries since 2009, the energy-saving focus of the Chinese manufacturing industry in the future should not be on the traditional high-energy-consuming industries, which have already achieved significant energy savings. Focusing on the energy consumption index of the manufacturing of smelting and pressing of nonferrous metals subindustry as an example, we use the grey forecasting GM (1, 1) model to forecast the index trend. We find that the index forecast line of this industry shows a continued downward trend before 2025. When compared with the forecast for the overall manufacturing industry, the outlook for the energy consumption of this subindustry, and thus of traditional high-energy-consuming industries, is more optimistic. That is, the small growth in the energy consumption indexes forecast for the whole manufacturing industry may not be caused by these energy-intensive industries. In other words, these industries will not "drag down" the energy intensity level of the whole manufacturing industry. In future, therefore, China's manufacturing industry should pay more attention to the energy-saving efforts of the industries outside of the traditional high-energy-consuming industries group. However, it is undeniable that the total energy consumption of the energy-intensive industries is a significant achievement, and these industries remain the major contributors to emission reductions [72].

*7.2. A Change Point in the Foreseeable Future Is Probable*

The GM (1, 1) model predicts that the manufacturing energy consumption index prior to 2025 will rise first and then decline. This indicates that China's manufacturing industry requires time to adjust its industrial structure and complete the transformation of the manufacturing industry. The establishment of green data centers and base stations is a strategic focus that aims to vigorously promote the low-carbon development of new materials, new energy, high-end equipment, and bio-industries [73]. Therefore, it is reasonable to believe that China's manufacturing industry could reach a new change point, after the initial slight increase in the energy consumption trend.

7.2.1. Although There Is a Downward Trend in the Index, Total Energy Consumption Remains High

From 1994 to 2015, the energy consumption index of the vast majority of manufacturing subindustries declined overall, indicating a significant improvement in energy efficiency per unit [17,18]. The change points of energy consumption indexes of most subindustries occurred around 2000 and 2008. However, the total energy consumption remains high and continues to increase.

7.2.2. There Is a Time Lag Effect for Energy Policies

Many previous studies have found that energy policies have had a positive effect on energy consumption [27,28,31–33]. This paper focuses on the characteristics of the annual rate of growth of the energy consumption index, which has received little attention in the previous literature, and finds that the effects of national energy policies have not been instantly reflected in the energy consumption index of the manufacturing industry. However, the annual rate of growth of the energy consumption index experiences an obvious decline within one to three years after the implementation of the central government's energy policies, which indicates that there is a short-term lag effect. This lag effect has been demonstrated in studies of carbon emission policies [74]. Once this lag is taken into account, the positive effects of implementing energy policies align with declines in the energy consumption index, which indicates that the government's policies to reduce energy consumption are scientific, feasible, and effective to a certain extent.

China's manufacturing industry needs to pay attention to the following points in terms of energy:

- The subindustries need to reduce their overdependence on importing core technology from abroad, and should vigorously strengthen independent innovation to further develop and use new energy sources with large reserves and little pollution, replace nonrenewable energy sources, and create more and more environmentally friendly new energy products.
- All of the subindustries should further phase out low-value-added industries that are high energy-consumers, while also increasing investments in energy saving by high-tech industries to improve the energy consumption structure, and strive to form an efficient, energy-saving development model.
- The government of China can establish an effective energy-saving incentive mechanism and supervise enterprises to ensure the effective implementation of the energy conservation law. The government should formulate policies for the long-term future to prevent enterprises from making decisions only in response to short-term inspections and taking actions that are not conducive to long-term sustainable development.

It should be noted here that this study has limits. First, the only way to obtain statistical data is through the National Bureau of Statistics of China. As publication of statistical data takes some time, the latest published data on the National Bureau of Statistics of China website are only up to 2015. As new data are made available, we will update our research. Second, this paper only studies manufacturing industry. The manufacturing industry is a typical industry in the industrial sector, which also includes other industries. Different industries have different characteristics. In order to make the results of the study universally meaningful, we will expand the sample size in the following

studies, and further study other industries in the industrial sector, make a comparative analysis of manufacturing and other industries.

**Author Contributions:** Conceptualization, W.S.; Methodology, W.S.; Formal analysis, Y.H. and L.G.; Data curation, Y.H. and L.G.; Writing-original draft preparation, W.S.; Writing-review & editing, W.S.

**Acknowledgments:** The work in this paper was supported by the National Natural Science Foundation of China (No. 61877034), the National Social Science Fund of China (No. 16ZDA047), the National Natural Science Foundation of China (No. 71673145), the Report Project on the Development of Philosophy and Social Sciences of China's Ministry of Education (No. 13JBG004).

**Conflicts of Interest:** The authors declare that we have no conflict of interest.

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
