# Peer review of "Analyzing and Forecasting Energy Consumption in China’s Manufacturing Industry and Its Subindustries"

_sustainability, doi:10.3390/su11010099_

Round 1

Reviewer 1 Report

The submitted manuscript evaluates the efficiency of energy use by the subindustries on China's manufacturing industry from 1994 to 2015. They calculate the energy consumption index of each industry in each year by using the method of maximum deviation. Then, they determine the years of change points in each industry using the method of grey relational analysis. The topic is interesting for this journal’s readers, but in this reviewer’s opinion, the paper lacks in clarity and this affects the technical content, which is not at the proper level for considering the paper to be published.

1- The description of methods (sections 3, 4) is based solely on formulas: please provide comments to better describe the proposed model.

2- Data used for the numerical validation are reported in Tables 1, 2: please provide some more details on these data and, for the sake of generality, please extend their range.

3- The simulation results do not show the advantages of the proposed method. Moreover, the quality of the figures are poor.

4- Please explain carefully Fig. 7.

5- Please provide simulation results to better describe the proposed method.
6- Algorithm flowchart?

7- In addition, references to prior works related to uncertain qualitative system could be improved by considering "Modified Differential Evolution Algorithm: A Novel Approach to Optimize the Operation of Hydrothermal Power Systems while Considering the Different Constraints and Valve Point ..." and " A Novel Algorithm for Optimal Operation of Hydrothermal Power Systems under Considering the Constraints in Transmission Networks" in energies, 2018.

Author Response

Dear Reviewer:
Thank you very much for your detailed and insightful comments.
In response to the your valuable comments on the manuscript (ID: sustainability-408107), some modifications have been made in this paper.
We have uploaded it as a PDF file.
Sincerely,

Wei Sun
Yufei Hou
Lanjiang Guo

Reviewer 2 Report

General comment: This paper explained the energy use of 27 subindustries in China's manufacturing industry and develops an energy consumption index for 1994–2015.

Introduction: Focus more on the objective of the paper, the importance of your study and results. 

Methodology: The database is old. Do you have recent data till 2017? Provide reasons for choosing those methods. Explain the limits and the advantages of the methods and provide more economic comments for introducing these methods. You have some panel data there? How do you make forecasts on such models? You should construct panel data models (dynamic panel data models?)?

Results: Compare your opinions with previous studies and highlight the novelty of your research.  Are the proposed models validated (errors homoskedasticity, normal distribution, independence checked)?  

Discussion: Interpretations of the results are not enough and a more critical position is required. Provide more economic comments for the results. State the limits of the research. How this research might continue?

Bibliography/References: up-date the list of references by new sources from WoS and SCOPUS databases

as a case

Khairalla, M.A.; Ning, X.; AL-Jallad, N.T.; El-Faroug, M.O. Short-Term Forecasting for Energy Consumption through Stacking Heterogeneous Ensemble Learning Model. Energies 2018, 11, 1605.

Kasperowicz R., Štreimikienė D. (2016), Economic growth and energy consumption:comparative analysis of V4 and the “old” EU countries, Journal of International Studies, Vol. 9, No 2, pp. 181-194. DOI: 10.14254/2071-8330.2016/9-2/14

Li, M.; Wang, W.; De, G.; Ji, X.; Tan, Z. Forecasting Carbon Emissions Related to Energy Consumption in Beijing-Tianjin-Hebei Region Based on Grey Prediction Theory and Extreme Learning Machine Optimized by Support Vector Machine Algorithm. Energies 2018, 11, 2475.

Brożyna J., Mentel G., Szetela B. (2016), Influence of double seasonality on economic forecasts on the example of energy demand, Journal of International Studies, Vol. 9, No 3, pp. 9-20. DOI: 10.14254/2071-8330.2016/9-3/1

Other remarks: The language could be improved. 

Author Response

(The authors gave the same response as above.)

Reviewer 3 Report

The paper entitled Analyzing and Forecasting Energy Consumption in China’s Manufacturing Industry and its Subindustries" is overall well written and interesting to readers. I would like to a critical description of the grey model itself: what are its major assumptions and limitations, advantages and disadvantages compared to other forecasting models? The other comments the authors must take into account are:

1. At the start of chapter 4 you take "The data in this paper are mainly obtained from the China Statistical Yearbook, the China Energy Statistics Yearbook, and the China Industrial Statistics Yearbook." Please add references for these sources.

2. Please comment the energy consumption indexes of the manufacturing subindustries given in table 2. Which tend to decrease/increase faster and which ones slower? Which subindustries are more energy intensive? 

3. The quality of figures 1-7. is poor and must be improved.

4. In table 5. you show change points for the 27 subindustries. Please comment on the possible relationship between the change point 2 and the global economic crisis 2008-2012. Also, why do you select three time periods and two change points for the time series? Why not 3 or 4 change ponts? Please comment.

Author Response

(The authors gave the same response as above.)

Reviewer 4 Report

The manuscript presents an analysis and forecasting of energy consumption for a number of Chinese manufacturing subindustries. The paper requires extra efforts to improve its quality and presentation for the prestigious journal Sustainability. A set of comments are expounded hereafter.

- The manuscript is well written and organized. However, there are some minor mistakes or improvements to make regarding the format of the document, as commented below.

The template of the Journal has to be used including the line numbering, which helps significantly to conduct a proper revision of the manuscript.

Between the last two authors’ names, “and” must be inserted.

In this reviewer’s humble opinion, a good practice for scientific text consists on capitalizing the first letter of the words that give place to an acronym.  

The grey model acronym, GM, should be defined within the Introduction, apart from the Abstract where is already properly defined.

Concerning the references, the names of the journals must be abbreviated, as indicated in the template. Most of them already are correct but some exceptions are the numbered as 3, 16, 24, 25, 44.

- About the content of the manuscript, the comments after a careful revision are the following:

In the keywords, an interesting term to be included is “manufacturing industry”, if the authors agree with the suggestion.

In the second section, the contextualization of the proposal is very well organized.

In the second section, or in the first one, according to the criteria of the authors, this reviewer misses a mention to the Made in China 2025 initiative. It is mentioned in the last paragraph of the sixth section (page 18) but given its importance in the industrial context, the manuscript should be include at least a brief mention in one of the two first sections.

A brief sentence to explain the content and/or structure of the third section would be desirable before the subsections that compose it. The same comment is applicable for the fourth and fifth sections.

Within the Methods section, it would be desirable to find some comment about the software environment that has been used by the authors. This information can be useful for the interested reader.

The quality of figure 1 should be improved. It is difficult to see the numbers. The same problem occurs for the rest of figures.

The selection of the change points seems adequate from this reviewer’s perspective.

The statement about the consideration of two categories in the subsection 5.2 seems to be incomplete. What are those categories?

The forecasting results are properly expounded.

As a conclusion of the revision, in its current state, the manuscript must address the provided suggestions to reach a better presentation and scientific level, according to the prestigious journal Sustainability.

Author Response

(The authors gave the same response as above.)

Round 2

Reviewer 1 Report

Congratulation!

Reviewer 2 Report

It is OK. The improvements were made.